# Intensive Care Risk Estimation in COVID-19 Pneumonia Based on Clinical and Imaging Parameters: Experiences from the Munich Cohort

**DOI:** 10.3390/jcm9051514

**Published:** 2020-05-18

**Authors:** Egon Burian, Friederike Jungmann, Georgios A. Kaissis, Fabian K. Lohöfer, Christoph D. Spinner, Tobias Lahmer, Matthias Treiber, Michael Dommasch, Gerhard Schneider, Fabian Geisler, Wolfgang Huber, Ulrike Protzer, Roland M. Schmid, Markus Schwaiger, Marcus R. Makowski, Rickmer F. Braren

**Affiliations:** 1Department of Diagnostic and Interventional Radiology, School of Medicine, Technical University of Munich, Ismaninger Str. 22, 81675 Munich, Germany; egon.burian@tum.de (E.B.); friederike.jungmann@tum.de (F.J.); g.kaissis@tum.de (G.A.K.); fabian.lohoefer@tum.de (F.K.L.); marcus.makowski@tum.de (M.R.M.); 2Department of Diagnostic and Interventional Neuroradiology, School of Medicine, Technical University of Munich, Ismaninger Str. 22, 81675 Munich, Germany; 3Department of Internal Medicine II, School of Medicine, Technical University of Munich, Ismaninger Str. 22, 81675 Munich, Germany; Christoph.spinner@tum.de (C.D.S.); tobias.lahmer@mri.tum.de (T.L.); Matthias.treiber@mri.tum.de (M.T.); fabian.geisler@tum.de (F.G.); wolfgang.huber@tum.de (W.H.); direktion.med2@mri.tum.de (R.M.S); 4Department of Internal Medicine I, School of Medicine, Technical University of Munich, Ismaninger Str. 22, 81675 Munich, Germany; Michael.dommasch@tum.de; 5Clinic for Anesthesiology and Intensive Care Medicine, School of Medicine, Technical University of Munich, Ismaninger Str. 22, 81675 Munich, Germany; Gerhard.schneider@tum.de; 6Institute of Virology, School of Medicine, Technical University of Munich, Ismaninger Str. 22, 81675 Munich, Germany; protzer@tum.de; 7School of Medicine, Dean, Technical University of Munich, Ismaninger Str. 22, 81675 Munich, Germany; markus.schwaiger@tum.de

**Keywords:** COVID-19, severe acute respiratory syndrome coronavirus 2 (SARS-CoV-2), clinical parameters, radiological parameters, computed tomography, intensive care unit

## Abstract

The evolving dynamics of coronavirus disease 2019 (COVID-19) and the increasing infection numbers require diagnostic tools to identify patients at high risk for a severe disease course. Here we evaluate clinical and imaging parameters for estimating the need of intensive care unit (ICU) treatment. We collected clinical, laboratory and imaging data from 65 patients with confirmed COVID-19 infection based on polymerase chain reaction (PCR) testing. Two radiologists evaluated the severity of findings in computed tomography (CT) images on a scale from 1 (no characteristic signs of COVID-19) to 5 (confluent ground glass opacities in over 50% of the lung parenchyma). The volume of affected lung was quantified using commercially available software. Machine learning modelling was performed to estimate the risk for ICU treatment. Patients with a severe course of COVID-19 had significantly increased interleukin (IL)-6, C-reactive protein (CRP), and leukocyte counts and significantly decreased lymphocyte counts. The radiological severity grading was significantly increased in ICU patients. Multivariate random forest modelling showed a mean ± standard deviation sensitivity, specificity and accuracy of 0.72 ± 0.1, 0.86 ± 0.16 and 0.80 ± 0.1 and a receiver operating characteristic-area under curve (ROC-AUC) of 0.79 ± 0.1. The need for ICU treatment is independently associated with affected lung volume, radiological severity score, CRP, and IL-6.

## 1. Introduction

At the end of 2019, infections with the novel severe acute respiratory syndrome coronavirus 2 (SARS-CoV-2) were first noted in Wuhan, China and rapidly spread to the rest of the world [1]. The lack of specific vaccines and treatments led to high infection and death counts around the world and thus caused a serious burden on national health care systems [2].

Even in highly developed health care systems, such as in Germany, with a very active testing and containment strategy alongside a federally guided preparation of the health care system for the coronavirus disease 2019 (COVID-19) pandemic, specific challenges are faced: pharyngeal swab-based polymerase chain reaction (PCR) testing is only highly sensitive during the early phase of infection and may miss the pulmonary phase of disease. Furthermore, limited resources both with respect to testing reagents and intensive care unit (ICU) capacities warrant identification of patients with primarily pulmonary disease manifestation at high risk for a severe course. Computed tomography (CT) was assigned a central role in patient stratification both in China and in Europe because of the high sensitivity, availability and speed [3,4,5]. Multiple studies have summarized clinical and CT imaging findings of COVID-19 pneumonia [6,7,8]. Typical imaging findings include confluent ground glass opacities, consolidation and crazy-paving patterns. More severe COVID-19 cases are frequently accompanied by laboratory and immunologic changes like C-reactive protein increases, lymphopenia and interleukin (IL)-6 peaks [9]. It stands to reason that the integration of imaging and clinical data can aid in the prediction of a severe course of disease in COVID-19 pneumonia and aid in patient stratification and triage, critical especially in an overloaded health care system as was observed in Italy [2] and more recently, New York [10].

Here we present the results of the integrated analysis of the first large scale outbreak in Southern Germany, Munich, exemplified by a cohort of 65 consecutive patients admitted to the emergency department and requiring hospitalization, and identify pertinent risk factors for ICU treatment.

## 2. Experimental Section

### 2.1. Patients

This study was conducted according to the principles set forward in the Declaration of Helsinki and according to Good Clinical Practice. All patients gave consent for scientific evaluation of clinical and imaging data at the time of admission. The local institutional review board of the Technical University of Munich has approved this prospective study (protocol numbers: 245/19 S-SR and 111/20 S). The study was designed as a retrospective cohort study. The STROBE checklist and patient recruitment flowchart (Appendix A) are included in the Appendix A [11].

Sixty-five consecutive patients were included in this study according to the Diagnosis and Treatment of Novel Coronavirus Pneumonia (5th version) of China [12], who presented in our hospital between March and April 2020. Inclusion criteria were defined as follows: Positive real-time reverse-transcriptase polymerase-chain-reaction (RT-PCR) testing for SARS-CoV-2 nucleic acid in throat swabs or lower respiratory tract lavage, characteristic imaging findings in thin-section CT, and moderate to severe symptoms (fever, dyspnea and/or dry cough) requiring hospitalization. Patients who developed one or more of the following symptoms were admitted to the ICU: respiratory rate ≥ 30 breaths per minute, peripheral resting state oxygen saturation ≤ 93%, invasively measured arterial oxygen tension (P_a_O_2_)/inspiratory oxygen fraction (F_i_O_2_) ≤ 300 mmHg (1 mmHg = 0.133 kPa), respiratory failure requiring mechanical ventilation, cardiovascular shock, and/or miscellaneous organ failure. Patients with a negative RT-PCR were excluded from the analyses.

The following patient medical record items were collected: comorbidities (systemic hypertension, diabetes mellitus, heart disease and chronic obstructive or restrictive pulmonary disease), medical history, and physical examination. Routine laboratory tests were performed including complete blood count and serum biochemistry (leukocytes, lymphocytes, creatinine, creatine kinase, creatine kinase myocardial band (CK-MB), C-reactive protein (CRP), troponin-T, lactate dehydrogenase (LDH), D-dimer, interleukin 6 (IL-6)).

### 2.2. MDCT Imaging

All patients were examined with the same 256-row multidetector computed tomography (MDCT) scanner (iCT, Philips Healthcare, Best, The Netherlands). Pulmonary MDCT was performed with the following parameters in full inspiration with arms elevated and no administration of contrast agent. One-hundred-and-twenty peak kilovoltage tube voltage, adapted tube load of averaged 200 mAs and minimum collimation (0.6 mm) were used. The mean dose length product was 217 ± 270 mGy/cm.

### 2.3. Severity Score in CT

Two radiologists with 8 (F.K.L.) and 3 years of experience (E.B.) performed qualitative image assessment blinded to the clinical data. For each of the 65 patients, chest CTs were evaluated according to the criteria presented in Figure 1. The spectrum of imaging manifestations ranged from grade 1 to 5. The intraclass correlation coefficient was calculated between the assessments.

### 2.4. Semi-Automated Lung Parenchyma Volume Quantification

CT datasets were semi-automatically segmented, and segmentations were manually corrected (IntelliSpace Portal 11, Philips Healthcare, Best, The Netherlands). Briefly, the complete lung volume and the volume of each lobe was quantified. Then, ground glass opacifications and consolidations were subtracted from the measured volume based on expert-determined threshold adaption, within the range of −600 HU to −770 HU (Figure 2 and Figure 3).

### 2.5. Statistical Analysis

Statistical analyses were performed in SPSS (version 26; SPSS Inc., Chicago, IL, USA) and Python 3.7.6. All tests were carried out using two-sided 0.05 level of significance. For multivariate random forest modelling, the proportion of ventilated lung parenchyma was computed for upper and lower lung from the semi-automated volume quantification. The middle lobe of the right lung was attributed to the upper lung. As the severity scores determined by the two radiologists had a high concordance, their average score was used in the model. All parameters were normalized to the unity interval. Single missing values were imputed using the mean value of continuous features or the mode in case of binary features. The random forest model was fit to all clinical and radiological data to predict the necessity of ICU treatment. Five-fold cross validation was performed with 80% of the data used for training and 20% for testing. Mean ± standard deviation (SD) importance for the features and the performance of the random forest model was assessed by cross validation.

## 3. Results

Sixty-five patients (23 female, 42 male, 61.5 ± 17.0 years) were included in this study. The most common symptoms were fever, cough, dyspnea, and gastrointestinal symptoms. The condition of 28 (6 female, 22 male, 64.9 ± 16.6 years, range 28–97 years) of these patients was deteriorating and necessitated ICU treatment. The clinical and laboratory parameters of the presented cohort are summarized in Table 1. The imaging-based severity assessment and volumetric lung parenchyma analysis is shown in Table 2. The analysis of the blood samples revealed a significant increase of leukocyte counts, CRP and IL-6 (*p* < 0.0001) and a significant reduction of the lymphocyte count in ICU patients (*p* < 0.0001) (Table 1).

The Cohen’s kappa testing showed excellent agreement in CT-based severity score rating between the two radiologists (κ = 0.81). Lung volume quantification revealed a significant increase in the percentage of opacifications in ICU patients (41.6%) compared to non-ICU patients (19.3%, t-test *p* < 0.001).

Five-fold cross validation of the Random Forest model yielded a mean ± SD sensitivity of 0.72 ± 0.1, specificity of 0.86 ± 0.16, accuracy of 0.80 ± 0.1 and ROC-AUC of 0.79 ± 0.1. The five most important features for classification were ventilation of the upper lung (0.184 ± 0.025), ventilation of the lower lung (0.123 ± 0.03), CRP (0.074 ± 0.018), radiological severity score (0.067 ± 0.011), and IL-6 (0.058 ± 0.006). The five most important features are displayed in Table 3. Imaging parameters alone resulted in a mean ± SD sensitivity of 0.78 ± 0.2, specificity of 0.78 ± 0.11, accuracy of 0.78 ± 0.1, and ROC-AUC of 0.79 ± 0.12. The addition of CRP to the imaging parameters resulted in a mean ± SD sensitivity of 0.76 ± 0.08, specificity of 0.81 ± 0.13, accuracy of 0.78 ± 0.06, and ROC-AUC of 0.79 ± 0.07. Finally, the addition of IL-6 to imaging parameters and CRP resulted in a mean ± SD sensitivity of 0.76 ± 0.08, specificity of 0.81 ± 0.13, accuracy of 0.78 ± 0.08, and ROC-AUC of 0.79 ± 0.07.

## 4. Discussion

We here present the multivariate analysis of clinical and imaging data of a cohort of 65 related to the first large scale outbreak of COVID-19 in Southern Germany. We identify imaging derived features (upper and lower lung opacifications, radiological severity estimation), CRP and IL-6 levels as the most important features for the prediction of the necessity of ICU admission. Our results underline the importance of the integration of CT imaging into the management of symptomatic COVID-19 pneumonia patients during this pandemic.

During the course of the COVID-19 pandemic, data related to epidemiological [13,14], clinical [15], virological [16], immunological [9], and imaging findings [6,17] of COVID-19, and their interdependence [18] is rapidly accumulating. In light of limited capacities and resources, with respect to ICU treatment, personal protective equipment and personnel [19], informed patient management is essential for planning. Even highly developed health care systems, such as the Italian, were faced with an unprecedented shortage of resources leading to the implementation of triage systems for the allocation of hospital and mechanical ventilation capacities [20].

Several current studies have correlated biomarkers and imaging derived markers with a severe disease course [21]. However, so far, to our knowledge, no studies have investigated markers for the prediction of the necessity of ICU treatment.

Our data indicate affection of the upper lung lobes as an easily identifiable and highly important predictive parameter of ICU treatment. This is consistent with the typical finding of COVID-19 pneumonia, which was often noted in the lower peripheral lobes of the lung, presumably involving the upper lobes in more advanced cases. This result, alongside the semi-quantitative severity score shows that simple metrics of pulmonary involvement can be used to predict clinical outcome. It remains unknown at this time how these imaging features relate to the individuals’ viral load. However, it is assumed that SARS CoV2 replicates in the lung tissue and it has been postulated that a failure to reduce viral load in the lung correlates with worse clinical outcome, which in turn is associated with more severe imaging findings [22,23]. Despite recent advances in molecular viral characterization [24], the recent evidence does not support a severity or mortality stratification based on viral load dynamics. However, the presented multiparametric severity assessment (including IL-6 and CRP) showed excellent correlation with disease severity and individuals’ risk for an aggravated course of disease.

Previous studies have addressed disease severity assessment in infectious lung diseases including qualitative and semi-quantitative approaches [25,26]. In these studies, imaging findings were also associated with higher mortality [25,26]. Our study confirms these findings and holds the additional benefit of a simplified image analysis and the integration of multiparametric patient data.

Although the presented data suggests the possibility of an estimation of disease severity in COVID-19 pneumonia using broadly available and simply assessed parameters, there are some limitations to our findings. The main limitation of our study is the lack of an external validation cohort. Further, the cohort size is small and therefore the importance of comorbidities is likely underrepresented in our analysis. Despite this, none of the ICU admitted patients seemed to suffer a clinical course defined by their comorbidity profile. Furthermore, in our current analysis we did not distinguish between different types of opacification (i.e., ground glass versus consolidated volumes) nor between clinically defined early and late stage manifestations of COVID-19 pneumonia. These may hold differential sensitivity for predicting the clinical course of disease and would first require validation in longitudinal imaging studies. Clearly, at this point we can only speculate on the clinical value of CT imaging for the prediction of individual viral load as other factors (e.g., host genetics and immune system interplay) may play in important role.

## 5. Conclusions

Our study presents CT imaging-derived findings and serum markers of inflammation as best suited predictors of the necessity for ICU admission of symptomatic COVID-19 pneumonia patients. We encourage the prospective validation of these results in clinical patient management.

## Figures and Tables

**Figure 1 jcm-09-01514-f001:**
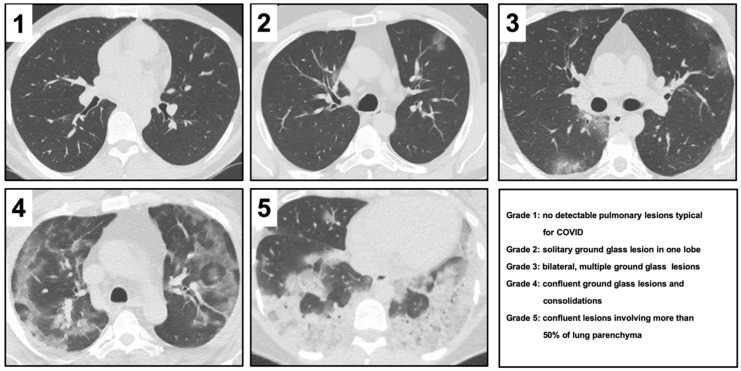
Images exemplifying the five radiological severity scores in computed tomography images alongside their descriptions. COVID, coronavirus disease.

**Figure 2 jcm-09-01514-f002:**
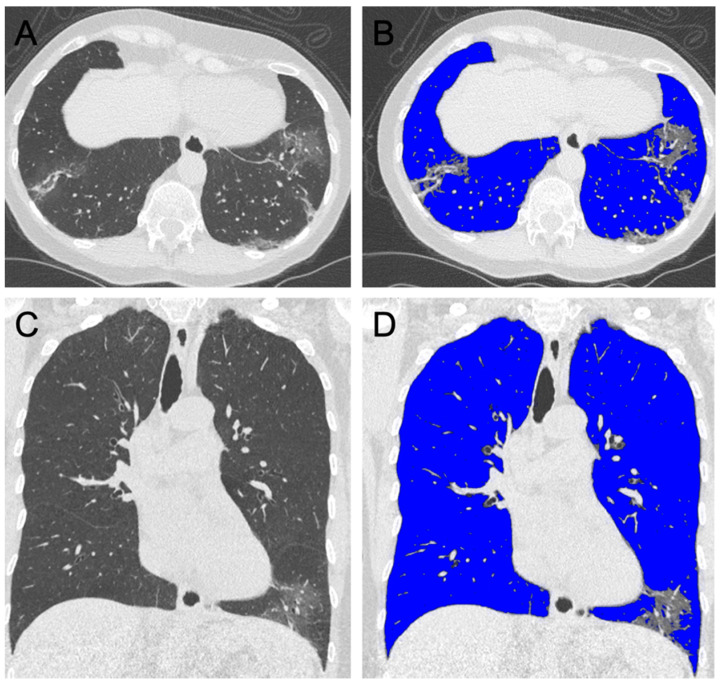
Axial (**A**,**B**) and coronal (**C**,**D**) reformations of the chest computed tomography (CT) of a 68-year-old female patient presenting with fever and cough. Radiological severity grade 3 was assigned. Lung volume quantification accounted for >90% ventilated volume (shaded blue in **B** and **D**).

**Figure 3 jcm-09-01514-f003:**
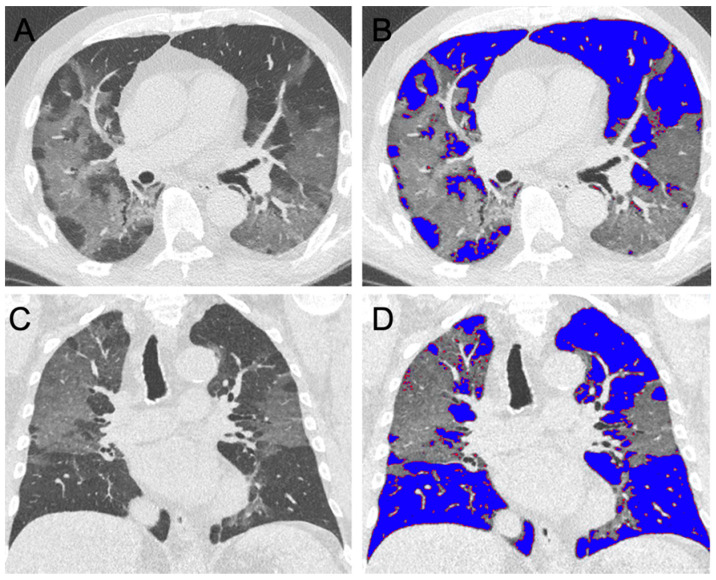
Axial (**A**,**B**) and coronal (**C**,**D**) reformations of the chest CT of a 78-year-old-male patient presenting with mild dyspnea. Radiological severity grade 4 was assigned. Forty-five percent of the lung volume is affected by characteristic parenchymal changes, most prominent in the right upper lobe (shaded blue in **B** and **D**).

**Table 1 jcm-09-01514-t001:** Descriptive statistics of the clinical and laboratory parameters. Parameters were compared between the two groups with student’s t test.

	Mitigated Group			ICU Group			
	*n*	Mean	SD	*n*	Mean	SD	*p*
Age (years)	37	59.0	17.1	28	64.9	16.6	0.170
Percutaneous oxygen saturation (%)	32	95.9	2.6	20	91.5	8.3	0.007
Temperature (°C)	37	37.8	0.9	21	37.6	17.9	0.173
Leucocytes (G/L)	37	5.6	2.5	28	8.2	4.0	0.002
Lymphocytes (%)	37	29.1	49.6	25	26.2	64.6	<0.001
CRP (mg/dL)	37	4.8	6.3	28	12.4	9.7	<0.001
Creatinine (mg/dL)	37	1.3	1.9	28	1.4	0.7	0.855
D-dimer (µg/mL)	31	1631.9	2573.3	18	2407.2	2856.5	0.334
LDH (U/L)	35	394.1	529.3	24	487.6	277.7	0.431
Creatine kinase (U/L)	34	153.91	165.2	24	632.4	1622.1	0.092
CK-MB (U/L)	1	37.0	32.4	5	42.8	25.3	
Troponin-T (ng/mL)	7	0.3	0.7	8	0.1	0.1	0.499
IL-6 (pg/mL)	25	51.7	65.6	12	103.9	43.6	0.017

ICU, intensive care unit; SD, standard deviation; CRP, C-reactive protein; LDH, lactate dehydrogenase; IL, interleukin; CK-MB, creatine kinase myocardial band.

**Table 2 jcm-09-01514-t002:** Descriptive statistics of the imaging parameters. Parameters were compared between the two groups with student’s t test.

	Mitigated Group			ICU Group			
	*n*	Mean	SD	*n*	Mean	SD	*p*
CT severity score	37	2.9	0.9	28	4.0	1.0	<0.001
Lung ventilated (%)	37	80.7	11.2	27	58.4	15.9	<0.001
Right lung (%)	37	80.7	10.7	27	57.0	17.6	<0.001
Right upper lobe (%)	37	83.1	12.4	27	56.4	21.7	<0.001
Right middle lobe (%)	37	88.0	7.4	27	71.0	17.4	<0.001
Right lower lobe (%)	37	74.5	13.9	27	49.4	20.2	<0.001
Left lung (%)	37	80.5	12.6	27	58.8	17.2	<0.001
Left upper lobe (%)	37	85.5	10.3	27	62.9	17.0	<0.001
Left lower lobe (%)	37	74.1	16.9	27	52.2	21.5	<0.001

CT, computed tomography.

**Table 3 jcm-09-01514-t003:** Mean ± SD of random forest classification feature importance across the cross-validation folds.

Feature	Mean Importance	Standard Deviation
Ventilation upper lung	0.184	0.025
Ventilation lower lung	0.123	0.030
CRP	0.074	0.018
Radiological Severity Score	0.067	0.011
IL-6	0.058	0.006

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
