# Peer review of "Intensive Care Risk Estimation in COVID-19 Pneumonia Based on Clinical and Imaging Parameters: Experiences from the Munich Cohort"

_jcm, 2020, doi:10.3390/jcm9051514_

Round 1

Reviewer 1 Report

Dear authors,

Your study is relevant and makes a relevant radiological observation in COVID-19 pneumonia.

As for the biomarkers, the relevance of IL-6 should be viewed with caution in the clinical context of screening patients who need ICU. It could exclude patients with perhaps less systemic severity but who need acute respiratory care.

As for CRP, it is a marker that will always be associated with others to be discriminatory between groups (ICU and non-ICU).

In general, I think your study brings the added value of joining the systematic analysis of the pulmonary image, in a way that can be innovative and simpler than other scores described.

The limitations mentioned will be overcome with a greater number of patients and you will certainly continue the work.
Congratulations!

Author Response

Dear Editor,

Dear Reviewers,

thank you very much for your effort and positive review provided for our submitted manuscript. Your comments and considerations, which we have responded to below, will substantially improve the quality of our study and manuscript.

Reviewer 1
Dear authors,

Your study is relevant and makes a relevant radiological observation in COVID-19 pneumonia.

Response: We would like to thank the reviewer for the overall positive assessment of our work. This will further encourage us to keep up the work on this highly interesting topic.

Comment 1:

As for the biomarkers, the relevance of IL-6 should be viewed with caution in the clinical context of screening patients who need ICU. It could exclude patients with perhaps less systemic severity but who need acute respiratory care. As for CRP, it is a marker that will always be associated with others to be discriminatory between groups (ICU and non-ICU).

Response:

We would like to thank the reviewer for this comment. We have now included a detailed comparison of the algorithms’ performance with and without CRP/IL-6 in the results section.

Reviewer 2 Report

Overall, this is an excellent effort to quantify imaging and relate them to outcomes. There are a few concerns I would like to raise.

Major comments:

Results
- Table 1 - the 0 and 1 groups should be named and listed in columns side-by-side to better see the difference (or lack thereof) in the various characteristics. I would consider separating radiological characteristics to a smaller, second table. These get buried in the table but are potentially of interest.

- lines 145-149 - because the outcome is ordinal and not continuous, ICC is not appropriate. I would consider using a different statistical test, such as Fleiss' kappa

- one key missing analysis is how much radiology can predict by itself, versus how much it benefits from adding CRP and IL-6. IL-6 is not always readily available. What are sens/spec/ROC with imaging alone? With inflammatory alone?

Discussion

- lines 181-189 - the focus on viral load is excessive. The focus should be on clinical outcomes. Viral load is a poor surrogate for severity, while imaging may not be. Would instead consider highlighting that viral load has not correlated well with mortality, but your imaging score has. 

- main limitation is lack of external validation with machine learning approach. This needs to be validated externally

- consider discussing other radiological severity scores, such as Grieser et al (Eur J Radiol 2012) or Sheshadri et al (PLOS One 2018) that have worked in other viral pneumonias. There should be some discussion on how yours differs or improves on these methods. 

Author Response

Dear Editor,

Dear Reviewers,

thank you very much for your effort and positive review provided for our submitted manuscript. Your comments and considerations, which we have responded to below, will substantially improve the quality of our study and manuscript.

Reviewer 2

Comment 2.1:
Table 1 - the 0 and 1 groups should be named and listed in columns side-by-side to better see the difference (or lack thereof) in the various characteristics. I would consider separating radiological characteristics to a smaller, second table. These get buried in the table but are potentially of interest.

Response 2.1:

We would like to thank the reviewer for this suggestion to improve the table clarity. We adapted the suggested changes and adjusted the tables accordingly.

Comment 2.2:

lines 145-149 - because the outcome is ordinal and not continuous, ICC is not appropriate. I would consider using a different statistical test, such as Fleiss' kappa.

Response 2.2:

As there are only two readers in our study Fleiss` Kappa does not apply in this setting. We calculated Cohen’s kappa instead.

Comment 2.3:

one key missing analysis is how much radiology can predict by itself, versus how much it benefits from adding CRP and IL-6. IL-6 is not always readily available. What are sens/spec/ROC with imaging alone? With inflammatory alone?

Response 2.3:

Thank you for this comment. Please also compare response to reviewer one. We have included the results of this analysis in the results section.

Comment 2.4:

lines 181-189 - the focus on viral load is excessive. The focus should be on clinical outcomes. Viral load is a poor surrogate for severity, while imaging may not be. Would instead consider highlighting that viral load has not correlated well with mortality, but your imaging score has.

Response 2.4:

We would like to thank the reviewer for this comment. The corresponding text passage has been revised in order to reduce the emphasis on viral load and to further strengthen the importance of imaging for the assessment of disease severity.

Comment 2.5:

main limitation is lack of external validation with machine learning approach. This needs to be validated externally

Response 2.5:

We agree with the reviewer’s comment. The limitation section was changed accordingly.

Comment 2.6:

consider discussing other radiological severity scores, such as Grieser et al (Eur J Radiol 2012) or Sheshadri et al (PLOS One 2018) that have worked in other viral pneumonias. There should be some discussion on how yours differs or improves on these methods. 

Response 2.6:

We would like to thank the reviewer for this suggestion. The manuscript will benefit from this comparison which was integrated into the discussion section.